# Uterine Aging and Reproduction: Dealing with a Puzzle Biologic Topic

**DOI:** 10.3390/ijms25010322

**Published:** 2023-12-25

**Authors:** Andrea Tinelli, Mladen Andjić, Andrea Morciano, Giovanni Pecorella, Antonio Malvasi, Antonio D’Amato, Radmila Sparić

**Affiliations:** 1Department of Obstetrics and Gynecology and CERICSAL (CEntro di RIcercaClinicoSALentino), “Veris delli Ponti Hospital”, 73020 Scorrano, LE, Italy; 2Clinic for Gynecology and Obstetrics, University Clinical Centre of Serbia, 11000 Belgrade, Serbia; andjicmladen94@gmail.com (M.A.); radmila@rcub.bg.ac.rs (R.S.); 3Department of Gynecology and Obstetrics, Pia Fondazione “Card. G. Panico”, 73039 Tricase, LE, Italy; drmorciano@gmail.com; 4Department of Obstetrics, Gynecology and Reproductive Medicine, Saarland University, 66421 Homburg, Saar, Germany; giovannipecorella@hotmail.it; 5Department of Biomedical Sciences and Human Oncology, University of Bari, 70121 Bari, BA, Italy; antoniomalvasi@gmail.com; 6Unit of Obstetrics and Gynecology, University of Bari, 70121 Bari, BA, Italy; antoniodamato19@libero.it; 7School of Medicine, University of Belgrade, 11080 Belgrade, Serbia

**Keywords:** uterine aging, reproduction, pregnancy, infertility, menopause, biological senescence

## Abstract

Uterine aging is the process of the senescence of uterine tissue, observed in all middle-aged mammals. Since the aging-related changes in the uterus are associated with infertility and poor pregnancy outcomes, with a lack of studies discussing uterine aging, authors reviewed uterine aging and its consequences on reproduction. MEDLINE, Scopus, and PubMed searches during the years 1990–2023 were performed using a combination of keywords and terms on such topics. According to the author’s evaluation, articles were identified, selected, and included in this narrative review. The aging process has an unfavorable impact on the uterus of mammals. There are different and selected molecular pathways related to uterine aging in humans and animals. Uterine aging impairs the function of the uterine myometrium, neurofibers of the human uterus, and human endometrium. These biological pathways modulate oxidative stress, anti-inflammatory response, inflammation, mitochondrial function, DNA damage repair, etc. All these dysregulations have a role in poorer reproductive performance and pregnancy outcomes in older mammals. The most recent data suggest that uterine aging is accompanied by genetic, epigenetic, metabolic, and immunological changes. Uterine aging has a negative impact on the reproductive performance in mammalian species, but it could be potentially modulated by pharmacological agents, such as quercetin and dasatinib.

## 1. Introduction

The physiological process of aging is thought to be regulated by various biological and genetic molecular processes. There has been a tendency for an increase in human life expectancy over the past few decades, but regrettably, this gain has not been followed by an increase in their health span [1]. The biological process of female reproductive aging, which includes changes to the reproductive system throughout a lifetime, is complex. Senescence, an organic and cellular process, is thought to play a role in women’s declining reproductive function because it is linked to infertility in women and poor pregnancy outcomes as they age [2].

Uterine aging is the process of the senescence of uterine tissue, which begins in all middle-aged mammals. The fall in fecundity has been reported to be partially caused by uterine aging, and several age-related changes in the hypothalamus, pituitary, and ovaries result in structural and functional alterations in the uterus [3]. The first studies about uterine aging and its negative consequences, especially on fertility, have been observed in patients submitted to oocyte donation and IVF programs [4,5]. The first study from the end of the XX century revealed there are uterine structural changes related to the injury in steroid synthesis [4,5]. All these uterine age-related processes led to poor reproductive outcomes in women with advancing age. Moreover, further research found that the uterine age-related process may also be to blame for a variety of undesirable pregnancy and delivery outcomes, including preterm birth, decreased uterine contractility, etc. [6].

There are contemporary studies that suggest that advanced maternal age is associated with unfavorable pregnancy outcomes [7]. Pregnants aged ≥ 45 years are associated with a higher cesarean delivery rate as well as with gestational diabetes mellitus, preeclampsia, placental abruption, placenta previa, postpartum hemorrhage, and preterm birth [7,8]. Additionally, the interesting data implicate there is a difference in the rates between maternal and neonatal negative outcomes between pregnancies achieved by artificial reproductive technologies [9].

Little research has addressed uterine aging and its effects, even though the ovary serves as the primary organ in the female reproductive system with both reproductive and endocrine functions [10]. Thus, this investigation reviewed the process of uterine aging and its health-related consequences, aiming to provide reproductive and biological information about uterine aging in animals and humans.

## 2. Results

### 2.1. Uterine Aging in Humans

Many studies on the human population investigated the impact of aging on the shape and function of uterine tissue as well as the various alterations that occur in the human uterus as we age. There are results about the influence of aging on the human uterine tissue. The observed finding suggests that uterine aging impairs the function of the uterine myometrium [11,12,13], neurofibers of the human uterus [14], and human endometrium [15,16,17,18,19,20].

The findings about the influence of the process of aging on the different components of the human uterus are shown in Table 1.

### 2.2. Uterine Aging in Animals

The studies conducted on the animals also revealed the influence of aging on the animal uterus. These are changes in the receptor expression, transcriptomic, and key molecular pathways in animal myometrium and endometrium [21,22,23,24,25,26,27], as well as the changes in the immune system and related components of the animal female reproductive tract [28,29,30,31,32,33]. The findings about the influence of the process of aging on the different components of the animal uterus are presented in Table 2.

## 3. Discussion

Senescent biological processes, which are associated with intricate alterations in the various organ systems during age, are what define the aging process [34] (Table 3).

The decline in reproductive performance and poor pregnancy outcomes in mammals is associated with advanced maternal age. Even though chromosome segregation mistakes in the oocyte during aging are associated with these unfavorable reproductive outcomes, recent research has shown that older moms, in both humans and mice, exhibit greater rates of infertility and pregnancy abnormalities in the absence of chromosome abnormalities. In animal studies, there were different placentation defects, more frequent in the population of older females [35]. All of these results point to a connection between the decline in reproductive abilities and uterine aging, not simply the aging of the ovaries.

The effects of age on the various uterine components in people and animals are not well studied, although aging’s impacts on the uterus often have a few essential characteristics.

### 3.1. Uterine Aging and Contractility

A decline in contractile uterine performance is generally reported during aging.

It could be explained by the influence of the different intracellular factors or signaling pathways that have a role in myometrial contractility, such as electrical conductance, intracellular calcium regulation, and mitochondrial function. These biological pathways and factors are all potentially compromised in older women [11]. There are findings about the influence of the aging process on some characteristics of the human myometrium. Crankshaw et al. [11] conducted a study aimed at examining the influence of maternal age on the contractility of human myometrium in pregnancy. The authors analyzed the number of contractile parameters such as maximal amplitude, mean contractile force, time to maximal amplitude, maximum rate of rise, and occurrence of simple and complex (biphasic and multiphasic) contractions during spontaneous and induced contraction. According to the results of the study, there is no significant correlation between maternal age and both spontaneous and induced contractions [11].

On the other hand, Du et al. [12] reported, in parturients older than 35 years, the changes in the SUR2B and Kir6.1 subunits in the KATP channels in the myometrial tissue. They revealed that the non-pregnant myometrium aging leads to the up-regulated SUR2B/Kir6.1 subunits of KATP channels. In the same study, authors showed the up-regulation of the SUR2B/Kir6.1 subunits of KATP channels.

Although controversial, these results suggested either no association in the proportion of biphasic or multiphasic contractions of human myometrium with maternal age or no impairment of uterine contractility and responsiveness of myometrium in vitro in the older mother. The aging-induced impairment of the expression of the potassium channel subunits, linked to the increasing age, could potentially explain the dysregulated uterine contractions during labor, increasing the need for cesarean section in older women [11,12].

### 3.2. Uterine Aging and Epigenetics

Maternal age is defined by chronological age. Thus, the epigenetic clock biomarkers cannot be usually used for the estimation of biological aging. It is crucial that neither uterine aging nor myometrial activities are visible as people age, both chronologically and epigenetically. The use of markers of epigenetic aging could be an informative indicator of advanced maternal age [13].

Erickson et al. [13] compared epigenetic age in blood and myometrial samples obtained by women during cesarean delivery at term gestation. The authors calculated epigenetic age using the Horvath, Hannum, GrimAge, and PhenoAge clocks. According to their results, markers of epigenetic aging, such as the expression of clock genes, could give information about the myometrial function and may be a possible predictor of different pregnancy-related conditions, like preeclampsia, and of the need for therapeutic strategies during pregnancy (diet/exercise, stress reduction, medical therapy, etc.) [13].

### 3.3. Uterine Aging and Neurotransmission

According to certain study results, the distribution of acetylcholine (AChE) and adrenaline neurofibers, which are linked to uterine arteries and myometrial smooth muscles in the uterine fundus, body, and cervix, decreases with age [14].

The human uterus is innervated by different sensory, parasympathetic, sympathetic, and peptidergic neuron fibers and many neurotransmitters, which have a role in reproductive physiology, all released after stimulation by adrenergic or cholinergic nerve fibers. Thus, there is currently debate in the literature regarding alterations to the parasympathetic and sympathetic nervous systems that occur with uterine aging. Age-related decreases in Adrenergic and AChE neurofibers have a detrimental effect on uterine physiology, particularly on the decreased likelihood of pregnancy success and concurrently increased risk of abortion and unfavorable obstetrical outcomes [14].

### 3.4. Uterine Aging and Human Endometrium

Changes in endometrial biology were observed during uterine aging. Age-related uterine changes have been observed either in the myometrium or in the endometrium [15]. After an analysis of expression and epigenetic changes in the human endometrial genes during uterine aging, it has been reported that about 81.48% of affected gene functions in the endometrium are linked to up-regulation of ciliary processes, with 91 genes related to cilia motility and ciliogenesis [15]. Moreover, there is the dysregulation of the vascular endothelial growth factor signaling pathway and inhibition of epithelial proliferation triggered and regulated by about 37 genes, with a role in cell cycle arrest, angiogenesis, insulin signaling, and telomere protection [15].

Endometrial dysregulation is mainly linked to the upregulation of the following genes involved in cilia motility: *DZIP1L*, *CUL9*, *CCDC13*, *CCDC113*, *TMEM107*, *TMEM67*, and *CEP162* [15]. Described up-regulation of ciliogenesis-related genes could be linked to down-regulation of proliferation of the epithelial cell because there is a reciprocal regulatory relationship between cilia formation and the cell cycle [36]. These results suggest that ovarian function is unaffected by aging. Nevertheless, there is also the endometrial aging paradigm in human reproduction, which may be connected to damaged embryo implantation. Additionally, considering the higher methylation of PTENP1 in older women and its tumor-suppressive role, it has been suggested that PTENP1 methylation is associated with subsequent elevation of its expression. It may also be a protective mechanism against the malignant transformation of endometrial tissue in older women [16].

The population of women who underwent the procedures of in vitro fertilization (IVF) and embryo transfer (ET) using donated oocytes was a good fit for the study of endometrial aging alterations. It has been reported a lower pregnancy rate in recipients of the oocyte donation who were ≥40 years old compared to younger recipients, <40 years of age, who were also in ovarian failure. The endometrial factor can be reassumed as follows: in 61% of the women in the group over 40, compared to just 29% in the group of women under 40, there was a failure to develop a critical endometrial thickness of 10 mm, as confirmed by ultrasonography [17].

In the endometrial age-related uterine changes, the important role is the age-related stem cell deficiency in the endometrium. It has been observed that the Sonic Hedgehog (SHH) signaling pathway activity in the endometrial stem cell declines during the aging process [18]. It has also been suggested that SERPINB2 is a master regulator of the SHH signaling pathway during the aging of the uterus [18]. It has been revealed that senescent stem endometrial cells potentially induce paracrine senescence in young counterparts via extracellular vesicles, cell contacts, and secreted factors [19,20]. The plasminogen activator inhibitor1 (PAI-1) and the insulin-like growth factor binding protein (IGFBP) have been suggested as the most prominent protein secreted by senescent endometrial stem cells [19,20].

### 3.5. Uterine Aging and Animal Experiments

In the animals, there are also reported changes in uterus tissue during aging.

It has been reported that there is an abnormal responsiveness of steroid hormones in the older ovariectomized mice than the younger [21,22,23,24,25,26,27,28,29,30,31,32,33]. In the older ovariectomized mice, there were lower pregnancy rates and a lower number of implantation sites than in younger mice who underwent embryo transfer [21].

Moreover, many genetic, transcriptomic, metabolic, and epigenetic changes are related to the aging process in the mice’s uterus. About 586 differentially expressed genes between the older and young mice have been observed [23]. The arachidonic acid metabolism and glutathione metabolism pathways are considered to play the main role in uterine cellular proliferation and decidualization during uterine aging [23]. It has been reported that older pregnant mice have a longer mean gestation, as well as a longer labor duration [24]. Patel et al. showed that the reductions in serum progesterone concentrations in older vs. younger animals were distinct, and cervical tissues in older mice were more distensible than in younger mice [24]. The authors observed, in the myometrium of older mice, either a decrease in the expression of oxytocin receptor and connexin-43 mRNA [24] or more frequent but shorter-duration spontaneous myometrial contractions and an attenuated response to oxytocin [24]. Finally, they reported a lower number of myometrial mitochondrial copy numbers in older than younger mice [24].

It has been shown that decreases in the concentration of the estradiol receptor concentration in the endometrium, endometrial stroma, myometrium, and epithelium of the mice during aging, with the decreased receptor content, per cell only, in the endometrial stroma [25].

Chong et al. [26] assessed the influence of aging, as well as the hormonal influence on the mouse uterus. They examined the transcript profile in myometrium obtained from four groups of virgin mice: (i) 10- to 12-week- and 28- to 30-week-old mice; (ii) 10- to 12-week- and 38- to 40-week-old mice; (iii) 38-week-old mice that had an ovariectomy or sham operation early in life; (iv) 38-week-old mice that had been treated with progesterone or vehicle containing implants from 8 to 36 weeks. In these animal models, they observed that the early ovariectomy prevented the age-related changes in the transcript profile of myometrium and that myometrial aging in mice is linked to the reproducible changes in the transcript profile, which could be prevented by interventions that inhibit cyclical changes in the female sex hormones [26].

In the mouse model, Sirtuin 1 is acknowledged as an important age-related regulator of uterine adaptability to pregnancy. The uterine-specific ablation of the Sirt1 gene leads to premature uterine aging and decreases in litter size from 1st pregnancy and becomes sterile (25.1 ± 2.5 weeks of age) after giving birth to the third litter [27]. The Sirt1 deficiency in mice led to the accelerated deposition of aging-related fibrillar Type I and III collagens in mice [24]. In addition, lower cell viability levels in endometrial cells have been reported, as well as a higher activation of “DNA damage checkpoint regulation” and the inhibition of “mitotic mechanisms” obtained from older cows compared to young [29].

In the uterus of aged mice (but also in humans), an over-activation of mTOR signaling, which is related to endometrial hyperplasia, was observed [30]. The uterine aging led to the downregulation of the *Pi3k/Akt1/mTOR* signaling pathway with the reduction in expression of related microRNAs miR34c, miR126a, and miR181b [31]. In the endometrial samples, gene expression of the proinflammatory cytokines Il17rb and chemokines Cxcl12 and Cxcl14 is higher in aged mice than in their young contrapuntal. These results suggest that these genes are a potential biomarker for endometrial aging and the prediction of age-related infertility [32].

### 3.6. Uterine Aging and Inflammation

There is a role for inflammation as low-grade, chronic, sterile inflammation associated with age-related disorders; it is assumed that dysregulations of uterine immune system cells contribute to older women’s reduced ability to reproduce [28,30,37].

Different signaling pathways that are involved in metabolism, tumorigenesis, epigenetic and transcriptomic regulation, and gene expression have been shown to be dysregulated during the aging process. This dysregulation may account for the higher incidence of uterine cancers and poorer reproductive outcomes in older humans and animals [38]. Considering the dysregulation in immune functions with body aging, it is expected there will be changes in the immune surveillance of the female reproductive tract.

Some researchers have carried out investigations on animals on the topic of inflammation and uterine aging.

Skulska et al. [28] reported a twofold increase in T cell number in aged mice. It has been reported that this impairment in the T cell number in the uterus of aged mice leads to the shift to the production of the proinflammatory type of cytokine in the uterine tissue of older mice [28].

In the bovine endometrial cells in vitro, obtained from older cows, it was observed that inflammation-related (predicted molecules are IL1A, C1Qs, DDX58, NFKB, and CCL5) and interferon-signaling (predicted molecules are IRFs, IFITs, STATs, and IFNs) pathways were more activated than in young cows [29].

The endocannabinoid system, which has a role in female fertility-related processes, is also modulated during reproductive aging. In fact, it has been shown in female mice during uterine aging a significant increase in TRPV1 (and transient receptor potential vanilloid type 1 channel) and enzymes NAPE-PLD, FAAH, and DAGL-β, all involved in the endocannabinoid system component metabolism. The age-dependent increases in TRPV1 and some endocannabinoid-related enzymes could be indicative of increased inflammation of the female mouse reproductive system during uterine aging [33].

Hence, aging is the most important factor in reproductive well-being, and the cellular senescent has a pivotal role in the uterine aging mechanisms. There are potential therapeutic interventions for these mechanisms, such as the modulators of oxidative stress, anti-inflammatory response, mitochondria, DNA damage, and signaling protein dysfunctions [39]. There are reports of senolytic drugs like quercetin and dasatinib slowing down the aging of human endometrial stromal cells and mousy uterine tissue [31,40]. Mitochondrial dysfunction has a role in the majority of mechanisms that lead to the aging process in the animal and human organic systems, and mitochondria are the primary endogenous source of reactive oxygen that contributes to the aging phenotype [41]. Taking into account, there are the multiple mechanisms that participate in uterine aging, such as oxidative stress, inflammation, fibrosis, DNA damage response, and cellular senescence, there are also some possible therapeutic approaches for uterine aging, such as herbal medicine products and stem cell therapy [42]. The application of the mesenchymal stem cells with antioxidative properties and also the stem cell conditioned medium, which contains paracrine factors, could be potential regenerative therapy for the aging uterus with lowered antioxidative capacity [43].

### 3.7. Uterine Aging and Pregnancy

Several studies have shown that older pregnant women than younger women experienced worse outcomes for both the mother and the fetus. It has been discovered that the rates of cesarean births and hospitalizations to the neonatal critical care unit—which are more common in the offspring of older mothers—rose in tandem with the ages of mothers [44].

It has been reported that nulliparous women aged ≥ 50 years are significantly more likely to undergo a trial of labor in comparison to women 20–34 years old. Additionally, they are more likely to experience an intrapartum cesarean delivery as well as severe maternal morbidity. Their infants are three times more likely to require neonatal intensive care unit admission. A similar result was observed among multiparous women older than 50 years when compared to the women 20–34 years old [45].

Research has demonstrated that hemorrhagic stroke is the leading cause of maternal mortality among mothers aged 40 and above, while preeclampsia is the leading cause of hemorrhagic stroke [46]. In addition, the likelihood of an uncomplicated pregnancy result, particularly a cesarean section and an early delivery, is linked to IVF and twin pregnancies in older women [47]. Among young pregnant women, older women who underwent artificial reproductive technology had a considerably increased risk of gestational hypertension, preeclampsia, and premature birth [48]. Moreover, the newborns in pregnancies made possible by artificial reproductive technology in older moms frequently have lower Apgar scores, are small for gestational age at birth, and have low birth weights [49]. They have more health problems during the first period of life than neonates of young mothers who underwent artificial reproductive technologies [46].

It is important to mention the pregnancy achieved using the process of oocyte donation, which has been improved over the last three decades [50]. Donating eggs has been associated with a higher risk of recurring miscarriages, infectious placental illnesses, preeclampsia, and postpartum hemorrhage, but it can also help people overcome physiological restrictions caused by insufficient ovarian reserve or unreliable gametes. These results suggest that whereas ovarian aging is the primary factor contributing to older women’s poor reproductive performance, uterine aging may also play a significant role in the poor reproductive and pregnancy outcomes experienced by patients who are older.

These findings highlight the need for global awareness of family planning choices, as well as for older mothers to receive care for themselves and their newborns. Furthermore, healthcare professionals, particularly obstetricians and perinatologists, must appropriately advise pregnant women who are older about all dangers and potentially bad outcomes for both mothers and neonates.

## 4. Materials and Methods

The authors investigated the available data concerning the biological process of uterine aging. Authors performed a selected PubMed, Medline, and Scopus research to reduce bias with other databases between 1990–2023, using a combination of keywords such as “uterus”, “aging”, “endometrium”, “myometrium”, “senescence”, “fertility”, “pregnancy”, “reproduction”, and “infertility”. The authors searched the database, selecting the clinical studies, the observational studies, and the basic and experimental studies. Once the articles were collected, the authors analyzed them according to the number of citations of each article, starting from the most cited to the least cited articles. Subsequently, we collected the data from each article and inserted them in the research paragraphs, summarizing the data collected. Peer-reviewed articles concerning uterine aging, as well as additional articles, were selected and identified from the references of relevant papers included in this paper.

## 5. Conclusions

The most recent information indicates that uterine aging is associated with genetic, epigenetic, metabolic, and immunological alterations despite the fact that there has been little research on uterine aging in the human population. These alterations may account for older women’s cancer genesis, difficult pregnancies associated with advanced maternal age, and poor reproductive outcomes. Future studies on the effects of aging on uterine tissue must consider and look into the effects of different senolytic agents on age-related changes and fertility in animals before extrapolating results to the general population since specific molecular pathways contribute to the aging process.

## Figures and Tables

**Table 1 ijms-25-00322-t001:** The influence of the process of aging on the different components of the human uterus.

Study	Main Findings
Crankshaw et al. [11]	There is no influence of the process of aging on uterine contractility.
Du et al. [12]	The uterine aging leads to an increase in the expression of the uterine SUR2B and Kir6.1 KATP channels.
Erickson et al. [13]	The uterine epigenetic clock correlates with the chronological maternal age.
Kosmas et al. [14]	The aging led to a decrease in the distribution of adrenergic and AChE neurofibers in the uterine fundus, body, and cervix.
Devesa-Peiro et al. [15]	The aging process reduces the gene expression related to cilia and aging hallmarks in the endometrium of women over 35 years old.
Kovalenko et al. [16]	The aging process is associated with the increase in PTENP1 methylation and its expression in endometrial tissue.
Check et al. [17]	Aging is related to the endometrium failure to generate a critical thickness.
Cho et al. [18]	Sonic hedgehog (SHH) signaling activity declines with aging, and SHH is a possible endogenous anti-aging factor in human endometrial stem cells.

**Table 2 ijms-25-00322-t002:** The influence of the process of aging on the different components of the animal uterus.

Study	Main Findings
Li et al. [22]	The aging led to an increase in Muc1 and PR levels and a decrease in Hand2 level endometrium of mice.
Kim et al. [23]	In silico analysis shows that the aging process in mice is associated with changes in expression in microRNAs-miR-223-3p, 155-5p, and 129-5p-within mice endometrium tissue.
Patel et al. [24]	The aging leads to a decrease in the expression of oxytocin receptor and connexin-43 mRNA as well as in the number of mitochondrial copy numbers in the myometrium of mice. Additionally, the aging leads to more frequent but shorter-duration spontaneous myometrial contractions and an attenuated response to oxytocin in mice.
Han et al. [25]	Aging is associated with the loss of estrogen receptors in the endometrial stroma of rats.
Chong et al. [26]	Aging is linked to the changes in the transcript profile of mouse myometrium. The regulator factor of interferon 7 has a possible role in the regulation of myometrial aging.
Cummings et al. [27]	The deletion of the SIRT1 gene led to the accelerated deposition of aging-related fibrillar Type I and III collagens in mice uteri.
Skulska et al. [28]	During the aging process, the profile of secreted inflammatory cytokines shifted toward the proinflammatory type in the murine reproductive epithelia.
Tanikawa et al. [29]	The aging is related to the activation of the inflammation-related (predicted molecules are IL1A, C1Qs, DDX58, NFKB, and CCL5) and interferon-signaling (predicted molecules are IRFs, IFITs, STATs, and IFNs) pathways in the cow endometrial cells as well as the activation of “DNA damage checkpoint regulation” and the inhibition of “mitotic mechanisms”.
Bajwa et al. [30]	There is an activation of the mTOR signaling in the aged women and mice.
Cavalcante et al. [31]	Aging is related to the downregulation of the Pi3k/Akt1/mTOR signaling pathway as well as a reduction in the expression of miR34c, miR126a, and miR181b.
Rossi et al. [33]	It has been shown a significant increase in components of the endocannabinoid system, such as TRPV1 receptor and enzymes NAPE-PLD, FAAH, and DAGL-β in the reproductive tract of female mice during aging.

**Table 3 ijms-25-00322-t003:** The main senescent biological processes influencing uterine aging [34].

Main Biological Process
Cellular Senescence
Chronic Inflammation
Genomic Instability
Epigenetic Alterations
Telomere Attrition
Disabled Macro Autophagy
Loss of Proteostasis
Mitochondrial Dysfunction
Dysbiosis

## Data Availability

No new data were created or analyzed in this study. Data sharing is not applicable to this article.

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
