# Peer review of "Uterine Aging and Reproduction: Dealing with a Puzzle Biologic Topic"

_ijms, 2023, doi:10.3390/ijms25010322_

Round 1

Reviewer 1 Report

Comments and Suggestions for Authors

Dear Authors,

Congratulations on the concept of creating a comprehensive literature review on uterine aging and its impact on perinatal outcomes. Thus far, the focus has been on the importance of ovarian aging, particularly the decline in egg quality with a woman's age, as a cause of reduced fertility and poorer perinatal outcomes. However, despite the numerous merits of the paper, several concerns arise:

Methodology – the presented manuscript lacks information on the criteria for selecting studies. The review includes publications on both human and animal subjects, and it seems that they should be separated because conclusions drawn from animal studies should not automatically be extrapolated to the human population. The keywords used for study selection are too broad, resulting in an analysis focused on the impact of a woman's age rather than the processes of uterine aging on fertility and perinatal outcomes. There is a lack of data on the number of selected studies or the percentage that were ultimately included. In my opinion, the criterion of citation count should not have been used, and the authors did not define the qualitative criteria for the cited studies.

The discussion, in my view, is somewhat chaotic and requires organization based on the type of information, such as studies on animals, molecular and genetic studies of human uteri, clinical studies discussing the influence of uterine changes against the background of systemic aging, hormonal and metabolic changes, and finally, differences in pregnancy and childbirth management in women for whom this may be their last chance to have a child.

Author Response

The presented manuscript lacks information on the criteria for selecting studies. The review includes publications on both human and animal subjects, and it seems that they should be separated because conclusions drawn from animal studies should not automatically be extrapolated to the human population.

Comments: though we conducted a narrative review, the reviewer is correct about the selection criteria, since there aren't many studies on this subject in the literature—roughly forty—and they're mostly focused on animals. Anyway, we addressed this matter in the review and made changes to the paper accordingly to reviewer’ comments.

The keywords used for study selection are too broad, resulting in an analysis focused on the impact of a woman's age rather than the processes of uterine aging on fertility and perinatal outcomes. There is a lack of data on the number of selected studies or the percentage that were ultimately included. In my opinion, the criterion of citation count should not have been used, and the authors did not define the qualitative criteria for the cited studies.

Comments: we partially agree with the reviewer, as out of 6 keywords chosen (uterine aging; reproduction; pregnancy; infertility; menopause; biological senescence), the first 4 are in line with the purpose of the manuscript, while the last 2 are on woman's aging (inserted only because they result in the cross-referencing of data in the search performed on the various databases). The manuscript is a narrative review, which does not require, from the point of view of editorial guidelines, specifications on the number of studies selected or the percentages of inclusion in the references, but we have also added this data in the revision of the manuscript, to meet the request of the reviewer.

The discussion, in my view, is somewhat chaotic and requires organization based on the type of information, such as studies on animals, molecular and genetic studies of human uteri, clinical studies discussing the influence of uterine changes against the background of systemic aging, hormonal and metabolic changes, and finally, differences in pregnancy and childbirth management in women for whom this may be their last chance to have a child.

Comments: we revised the discussion, editing the text and improving its understanding, trying to make the discussion more fluid. We divided the discussion into paragraphs, but we have tried to make the exposition clearer and more organized.

Reviewer 2 Report

Comments and Suggestions for Authors

Good review of the literature on the aging of the uterus and its processes in women. The comparison with the (little) data we have on animals is excellent and the evaluation of genetic alterations is excellent. 

The main question that the paper aims to answer is the evaluation of the factors that contribute to uterine aging, with a perfect review of the data in the literature

Although the topic is not completely original, it is the review of all the papers on the biological factors involved in the aging of the myometrium and endometrium and the comparison between women and animals that makes the paper original. Being a review, the methodology applied is correct and complete and the evaluation of the literature makes the study useful

In this case the tables are the list of publications that deal with each topic evaluated and are complete and exhaustive

Line 84: replace “inanimal myometrium” whit “in animal myometrium”

Line 109: remove “)”

Author Response

Good review of the literature on the aging of the uterus and its processes in women. The comparison with the (little) data we have on animals is excellent and the evaluation of genetic alterations is excellent. 

The main question that the paper aims to answer is the evaluation of the factors that contribute to uterine aging, with a perfect review of the data in the literature

Although the topic is not completely original, it is the review of all the papers on the biological factors involved in the aging of the myometrium and endometrium and the comparison between women and animals that makes the paper original. Being a review, the methodology applied is correct and complete and the evaluation of the literature makes the study useful. In this case the tables are the list of publications that deal with each topic evaluated and are complete and exhaustive

Line 84: replace “inanimal myometrium” whit “in animal myometrium”

Line 109: remove “)”

Comments: we have modified the text by correcting the errors reported by the reviewer.

Reviewer 3 Report

Comments and Suggestions for Authors

Interesting review on uterine aging. However, few more thigs should be further elaborated and adressed: 1. role of oxidative stress and DNA damage response

2. potential therapy that is described like stem cell therapy, herbal therapy etc. as protective therapy against uterine aging. It is important to understand processes and mechanisms associated with uterine aging and to improve fecundity and reproductive outcome in women of more advanced reproductive age.

Comments on the Quality of English Language

English fine

Author Response

Interesting review on uterine aging. However, few more thigs should be further elaborated and addressed:

  1. role of oxidative stress and DNA damage response.
  2. potential therapy that is described like stem cell therapy, herbal therapy etc. as protective therapy against uterine aging. It is important to understand processes and mechanisms associated with uterine aging and to improve fecundity and reproductive outcome in women of more advanced reproductive age.

Comments: we have added, according to what is available in the literature on uterine ageing, the two topics in the manuscript.

Round 2

Reviewer 3 Report

Comments and Suggestions for Authors

This is an interesting review on uterine aging background. Authors have improved previous version.